# Physiological, Photosynthetic Characteristic and Transcriptome Analysis of *PsnWRKY70* Transgenic *Populus simonii* × *Populus nigra* Under Salt Stress

**DOI:** 10.3390/ijms26010081

**Published:** 2024-12-25

**Authors:** Hui Zhao, Wenhu Wang, Yujie Fan, Guifeng Liu, Shaokang Guo, Guoqiang Fan

**Affiliations:** 1College of Forestry, Henan Agricultural University, Zhengzhou 450002, China; 2Institute of Paulownia, Henan Agricultural University, Zhengzhou 450002, China; zhao_hui_zh@163.com (H.Z.); wwh15693348309@126.com (W.W.); fanyujie@henau.edu.cn (Y.F.); guoshaokang17@163.com (S.G.); 3State Key Laboratory of Tree Genetics and Breeding, Northeast Forestry University, Harbin 150040, China

**Keywords:** *PsnWRKY70*, *Populus simonii* × *Populus nigra*, physiological and photosynthetic characteristics, RNA-seq, salt stress

## Abstract

The *Psn*WRKY70 transcription factor (TF) was reported to play an important role in the salt stress response mechanism of *Populus simonii × Populus nigra* in our previous research, and we also produced several *PsnWRKY70* overexpression (OEXs) and RNAi suppression (REXs) *P. simonii* × *P. nigra* lines. In order to further compare the photosynthetic and physiological characteristics of NT (non-transgenic line) and transgenic lines under salt stress, the dynamic phenotypic change, Na^+^ and K^+^ content in leaf and root tissues, superoxide dismutase (SOD) and peroxidase (POD) activity, malondialdehyde (MDA) content, chlorophyll content (Chl), photosynthesis parameters (net photosynthetic rate, P_n_; stomatal conductance, Gs; intercellular CO_2_ concentration, C_i_; transpiration rate, T_r_), chlorophyll fluorescence parameters (electron transport rate, ETR; maximum photochemical efficiency of photosystem II (PSII), F_v_/F_m_; actual efficiency of PSII, Φ_PSII_; photochemical quenching coefficient, q_P_; non-photochemical quenching, NPQ; the photosynthetic light-response curves of Φ_PSII_ and ETR) and RNA-seq of NT, OEX and REX lines were detected and analyzed. The phenotypic observation, MDA content and Chl detection results indicate that the stress damage of REXs was less severe than that of NT and OEX lines under salt stress. Photosynthesis parameter (P_n_, Gs, T_r_ and C_i_) and chlorophyll fluorescence parameter (ETR, F_v_/F_m_, Φ_PSII_ q_P_ and NPQ) detection results indicate that the REX lines exhibited much better photosynthetic adaptability than NT and OEX lines during salt stress. The photosynthetic light-response curves of Φ_PSII_ and ETR of NT, OEX and REX lines indicate that REXs exhibited better ability to activate the photosynthetic protection mechanism and adapt to a certain degree of strong light than NT and OEX lines under salt stress. RNA-seq analysis indicates that the DEGs between OEX1 vs. NT and REX1 vs. NT in different tissues (apical bud and fifth functional leaf) were all different in category and change trend. The expression of *PsnWRKY70* was significantly up-regulated in both the apical bud and fifth functional leaf of OEX1, and showed no significant change (namely maintained low expression level) in both the apical bud and fifth functional leaf of REX1, thus indicating the negative regulation role of *PsnWRKY70* in *P. simonii* × *P. nigra* under salt stress. Additionally, there were a lot of stress response-related TF genes (such as *bHLH*, *WRKY*, *MYB*, *NAM* and *AP2/EREBP*) and photosynthesis-related genes among all the DEGs. In REX1, the expression of three *Photosystem I P700 chlorophyll a apoprotein A1* genes (Potri.003G065200, Potri.013G141800 and Potri.019G028100) and a *Photosystem II protein D1* gene (Potri.013G138300) were significantly up-regulated after 6 days of salt stress. In OEX1, the *Heterodimeric geranylgeranyl pyrophosphate synthase small subunit* gene (Potri.015G043400) and *Phospho-2-dehydro-3-deoxyheptonate aldolase 1* gene (Potri.007G095700) were significantly down-regulated after 6 days of salt stress. These photosynthesis-related genes are probably regulated by *Psn*WRKY70 TF in response to salt stress. In conclusion, the REX lines suffered less severe salt damage and exhibited better photosynthetic adaptability than NT and OEXs under salt stress. The differences among the DEGs between OEX1 vs. NT and REX1 vs. NT in apical bud and fifth functional leaf, and the significantly differentially expressed photosynthesis-related genes are probably the key clues for discovering the photosynthesis adaptability mechanism of *PsnWRKY70* transgenic *P. simonii* × *P. nigra* under salt stress.

## 1. Introduction

With the development of the modern industrial society, environmental problems are more and more severe. Adverse environmental stress factors prevent plants from realizing their full genetic potential [1]. Among the various abiotic environmental stress factors, soil salinization is a specifically serious global problem. High salinity negatively affects growth and development of plants: on the one hand, it imposes cellular osmotic stress by reducing the soil water potential, on the other hand, it causes excessive ion (particularly Na^+^ and Cl^−^) uptake that ultimately interferes with various metabolic processes. Salt tolerant responses result in adaptation for growth continuation, though at a modified rate, to adaptation for survival where the growth is almost suspended but the plant stays alive until the stress is relieved, while salt-sensitive responses could result in accelerated senescence and eventually death of plant. Generally speaking, when plants are confronted with high salinity conditions, they initiate complicated gene regulatory networks (signal transduction) and metabolic processes in response to the salt stress [2,3].

On the molecular level, plant salt stress signal transduction consists of ionic and osmotic homeostasis signaling pathways, detoxification (i.e., damage control and repair) response pathways and growth regulation pathways involving several signaling factors and a large number of stress-related genes. And these salt stress-responsive genes are reportedly modulated by some specific types of transcription factors (TFs) such as bZIP, MYB, ERF/AP2, NAC and WRKY [4,5,6,7,8,9].

Among these salt stress-responsive TFs, WRKY TFs are encoded by a large gene superfamily in higher plants. For example, *Arabidopsis thaliana*, rice (*Oryza sativa* subsp. Japonica), soybean (*Glycine max*) and poplar (*Populus trichocarpa*) contain a large number of *WRKY* genes in their genomes [10,11,12,13,14]. The WRKY proteins owe their names to the highly conserved 60-amino-acid WRKY domains, which contain a core conserved motif (WRKYGQK) at N-termini and a novel zinc-finger-like motif at C-termini [15]. According to the WRKY domain and the zinc-finger-like motif type, the WRKY superfamily can be classified into three groups, namely Group I, Group II and Group III. Generally speaking, Group I members contain two WRKY domains, whereas the Group II and Group III members contain one, and the main difference between Group II and Group III is that Group II members contain a Cx4–5Cx22–23HxH zinc finger-like motif, while Group III members contain the Cx7Cx23HxC. Additionally, further phylogenetic studies have divided Group II into five subgroups (i.e., Groups IIa, IIb, IIc, IId, and IIe) [4,16]. The conserved WRKY domain and its corresponding *cis*-regulatory element, W-box (T**TGAC**C/T), act as molecular switches for target gene expression, regulating their temporal and spatial expression [15,17,18,19].

*WRKY* gene family members have been reported to play crucial roles in the regulation of transcriptional reprogramming associated with various biological processes such as development, senescence, seed dormancy and germination, biotic stress responses, and abiotic stress responses, and most notable reports are all about stress response processes [20,21,22,23,24,25,26]. The majority of earlier studies have concerned the vital roles that numerous WRKY TFs played in plant immune response processes [17,27,28,29], whereas recent studies have begun to explore the biological functions of WRKYs in abiotic stress (high salinity, drought, cold, heat, etc.) response signal transduction networks [11,30,31,32,33,34,35,36]. For instance, the ectopic expression of wheat (*Triticum aestivum* L.cv. Chinese Spring) *WRKY44* gene enhanced drought, salt and osmotic stress tolerance of transgenic tobacco [37]. Overexpression of cotton (*Gossypium hirsutum*) *WRKY34* gene in *Arabidopsis* enhanced salt tolerance of the transgenic plants [38]. The *CsWRKY46* gene from cucumber (*Cucumis sativus* var. *sativus* L.) was also reported to confer cold tolerance to transgenic plants by regulating a set of cold stress-responsive genes in an abscisic acid-dependent manner [39].

On the physiological and biochemical levels, plants respond to salt stress by continually adjusting key metabolic processes, such as photosynthesis [40,41], energy metabolism [42], and molecular synthesis [43]. As one of the most important processes occuring in the chloroplasts of higher plants, photosynthesis accumulates organic metabolites by absorbing and converting light (solar) energy and CO_2_ into chemical energy and O_2_ [44]. When plants are subjected to salt stress (high NaCl concentration treatment), the chloroplast structures are affected, the chlorophyll content is decreased, and the net photosynthetic rate is also reduced [45].

As a native tree species that is distributed widely from Heilongjiang River to Yellow River in northern China, *Populus simonii* × *Populus nigra* (*P. simonii* × *P. nigra*) has excellent physio-ecological characteristics, and is therefore well adapted to cold, drought, high-salinity and barren conditions. This hybrid poplar was widely used as a valuable resource for stress-resistant breeding in woody plants [46,47]. According to our previous transcriptome analysis and qRT-PCR (quantitative real-time PCR) experiments, the expression level of a *P. simonii* × *P. nigra WRKY* gene (named as *PsnWRKY70*, POPTR_0016s14490) was significantly down-regulated by salt stress [48]. Then, we performed genetic transformation to generate *PsnWRKY70*-overexpressed (named as OEX) and *PsnWRKY70*-repressed (named as REX) *P. simonii* × *P. nigra* lines. The phenotype investigation and physiological/biochemical indices assays on non-transgenic (NT) and transgenic lines after salt stress treatment (120 mM NaCl) suggest that the REX lines exhibited stronger salt tolerance than NT, whereas the OEX lines did not display obvious advantages under salt stress [49].

In the present study, we detected the Na^+^ and K^+^ content, superoxide dismutase (SOD) and peroxidase (POD) activity, malondialdehyde (MDA) content, chlorophyll content, photosynthesis parameters (net photosynthetic rate (P_n_), stomatal conductance (G_s_), transpiration rate (T_r_) and intercellular CO_2_ concentration (C_i_)), and chlorophyll fluorescence parameters (maximum photochemical efficiency of photosystem II (PSII) (F_v_/F_m_), actual efficiency of PSII (Φ_PSII_), electron transport rate (ETR), photochemical quenching coefficient (q_P_), non-photochemical quenching (NPQ)) of NT, OEX and REX lines under salt stress. Simultaneously, RNA-seq of NT, OEX1 and REX1 lines was performed 6 d after salt stress treatment. Based on the above experiments, we attempted to compare and analyze the physiological and photosynthetic characteristics of non-transgenic and *PsnWRKY70* transgenic *P. simonii* × *P. nigra* under salt stress, and further excavate the photosynthesis-related genes and pathways that are directly or indirectly regulated by *Psn*WRKY70 TF during salt stress response in *P. simonii* × *P. nigra*.

## 2. Results

### 2.1. Phenotypic Changes in NT, OEX1 and REX1 Lines During Salt Stress

Significant phenotypic differences were found among the three lines under the same growth conditions before salt stress treatment. Compared to NT plants, REX1 plants were taller and had a thicker diameter, whereas OEX1 plants were shorter and had a thinner diameter (Figure 1A). On the third day after salt stress, the leaves of NT and OEX1 began to yellow, while the leaves of REX1 were still green (Figure 1B). After 15 days of salt stress treatment and 6 days of re-watering, NT plants were found to be in a poor state, showing sparse and gradually yellowing leaves, dead terminal buds, but growing axillary buds. REX1 plants maintained sparse but green leaves, with surviving terminal buds and a large number of axillary buds after re-watering. OEX1 plants were in the worst state, with sparse yellow leaves, dead terminal buds, and almost no axillary buds growing after re-watering (Figure 1C). The above results indicate that the suppression of *PsnWRKY70* enhances the salt tolerance of *P. simonii* × *P. nigra*, while the overexpression of *PsnWRKY70* has the opposite effect.

### 2.2. Na^+^, K^+^ Content, Antioxidase Activity and Membrane Lipid Damage of NT, REX and OEX Lines Under Salt Stress

Under normal growth conditions, the Na^+^ and K^+^ content exhibited no obvious difference among NT, OEX1 and REX1 lines both in leaf and root, and the Na^+^/K^+^ maintained at a low level. After 6 days of salt stress treatment, the Na^+^ content of NT and transgenic lines all significantly increased, while the K^+^ content of NT and transgenic lines all significantly decreased, both in leaf and root tissues. The Na^+^ content increase rate of OEX1 was larger than that of NT and REX1, and the K^+^ content decrease rate of OEX1 was also larger than that of NT and REX1 (Figure 2A,B). The Na^+^/K^+^ of NT, OEX1 and REX1 all significantly increased both in leaf and root, and the Na^+^/K^+^ of OEX1 (2.81 ± 0.52 in leaf; 4.01 ± 0.34 in root) was much larger than that of NT (1.55 ± 0.06 in leaf; 2.54 ± 0.18 in root) and REX1 (0.78 ± 0.02 in leaf; 0.96 ± 0.03 in root) (Figure 2C).

SOD and POD activity of NT and transgenic lines were all significantly increased on the third day after salt stress. The SOD activity increase rate of NT and REX1 was 117.57% and 200.66%, while that of OEX1 was 50.36%. The POD activity increase rate of NT and REX1 was 56.79% and 135.73%, while that of OEX1 was 19.26%. Compared with NT, the SOD and POD activity increase rate of REXs was more obvious, while that of OEXs exhibited the opposite case (Figure 2D,E). This indicates that REXs have stronger ability to induce antioxidase in response to salt stress.

MDA content of NT and transgenic lines were also increased on the third day after salt stress. The MDA content increase rate of NT and REX1 was 35.46% and 8.29%, while that of OEX1 was 96.94% (Figure 2F). This indicates that REXs suffered less membrane lipid damage than NT and OEXs under salt stress.

### 2.3. Chlorophyll Content and CCI Value of NT, REX and OEX Lines Under Salt Stress

After the salt stress treatment with 120 mM NaCl, the chlorophyll content (Chl, mg·g^−1^) of all lines showed a decreasing trend with the extension of stress duration. Compared with NT and REXs (namely REX1, REX2 and REX3) lines, Chl of OEXs (namely OEX1, OEX2 and OEX3) lines began to decline rapidly just 3 days after salt stress. On the ninth day after salt stress treatment, Chl of NT lines decreased by 50.27%, Chl of OEX1, OEX2 and OEX3 decreased by 52.63%, 49.39% and 48.89%, respectively, Chl of REX1, REX2 and REX3 decreased by 21.89%, 27.73% and 31.21%, respectively (Figure 3A).

CCI value of all lines also showed a decreasing trend with the extension of stress time. Compared to NT lines, the CCI of REX lines decreased slower, whereas OEX lines decreased faster during salt stress treatment. On the ninth day after salt stress, the CCI of NT lines decreased by 33.33%, OEX1, OEX2, and OEX3 lines decreased by 41.99%, 38.27%, and 40.16%, respectively, REX1, REX2, and REX3 lines decreased by 20.00%, 10.00%, and 16.43%, respectively (Figure 3B). These data indicate that the suppression of *PsnWRKY70* can assuage the chlorophyll content decline of *P. simonii* × *P. nigra* during salt stress.

### 2.4. Photosynthesis Parameters of NT, REX and OEX Lines Under Salt Stress

The net photosynthetic rate (P_n_, μmol·m^−2^·s^−1^) of all lines showed a decreasing trend with the extension of salt stress treatment time. The P_n_ of NT and REX lines did not decreased significantly, while that of OEX lines decreased rapidly just on the first day after stress treatment. And the P_n_ of OEX lines remained lower than that of REX lines. On the ninth day after salt stress, P_n_ of NT decreased by 77.52%, OEX1, OEX2 and OEX3 decreased by 86.72%, 82.35% and 80.89%, respectively, REX1, REX2 and REX3 decreased by 72.48%, 76.49% and 78.14%, respectively (Figure 4A), indicating that the suppression of *PsnWRKY70* can slow down the P_n_ decline of *P. simonii* × *P. nigra* under salt stress and maintain better photosynthesis ability.

With the extension of salt stress duration, the intercellular CO_2_ concentration (C_i_, μmol·m^−2^·s^−1^) of NT, OEX and REX lines first decreased and then increased. C_i_ of NT and OEX lines reached the minimum value on the third day after salt stress, with NT lines decreased by 28.51%, while OEX1, OEX2 and OEX3 decreased by 35.74%, 31.84% and 31.61%, respectively. C_i_ of REX1, REX2 and REX3 reached the minimum value on the sixth day after salt stress, decreased by 21.80%, 23.13% and 25.35%, respectively (Figure 4B).

The stomatal conductance (G_s_, mol·m^−2^·s^−1^) and transpiration rate (T_r_, mol·m^−2^·s^−1^) of all lines showed a decreasing trend with the extension of stress duration. On the ninth day after stress treatment, G_s_ of NT decreased by 57.80%, while REX1, REX2 and REX3 decreased by 42.06%, 43.81% and 44.76%, respectively, and OEX1, OEX2 and OEX3 decreased by 69.45%, 60.38% and 60.00%, respectively (Figure 4C). T_r_ of NT and OEX lines declined rapidly on the first day after stress treatment, while that of REX lines began to decline rapidly on the third day after treatment. After 9 days of salt stress, T_r_ of NT decreased by 66.08%, REX1, REX2 and REX3 decreased by 60.56%, 62.29% and 70.49%, respectively, and OEX1, OEX2 and OEX3 decreased by 84.66%, 79.66% and 71.41%, respectively (Figure 4D). These results indicate that suppression of *PsnWRKY70* can limit stomatal conductance, thereby reduce transpiration rate and maintain cell permeability, under salt stress.

### 2.5. Chlorophyll Fluorescence Parameters of NT, REXs and NEXs Lines Under Salt Stress

After 9 days of salt stress, the photosynthetic electron transport rate (ETR, μmol·m^−2^·s^−1^) of all lines significantly decreased (with different falling ranges). The ETR of NT plants decreased by 41.30%, that of OEX1, OEX2 and OEX3 lines decreased by 64.49%, 55.16% and 52.31%, respectively, and that of REX1, REX2 and REX3 lines decreased by 29.60%, 35.64% and 37.55%, respectively. The ETR of REX lines was significantly higher than that of NT and OEX lines, i.e., REXs > NT > OEXs (*p* ≤ 0.05) (Figure 5A).

On the ninth day after salt stress treatment, F_v_/F_m_ of all lines showed different degrees of decline: F_v_/F_m_ of NT decreased by 48.83%, and that of OEX1, OEX2 and OEX3 lines decreased by 75.32%, 71.27% and 72.69%, respectively, that of REX1, REX2 and REX3 lines decreased by 32.89%, 37.05% and 38.28%, respectively. F_v_/F_m_ of REX lines was significantly higher than that of NT and OEX lines, i.e., REXs > NT > OEXs (*p* ≤ 0.05) (Figure 5B).

After 9 days of salt stress, the change trends of actual photochemical efficiency (Φ_PSII_) and photochemical quenching coefficient (q_P_) of all lines were similar to that of ETR, and the overall decrease range was OEX >NT >REX. The Φ_PSII_ and q_P_ of REX lines were significantly higher than those of NT and OEX lines, i.e., REXs > NT > OEXs (*p* ≤ 0.05) (Figure 5C,D).

After 9 days of salt stress, the non-photochemical quenching coefficient (NPQ) of NT increased by 137.11%, NPQ of OEX1, OEX2 and OEX3 lines increased by 182.33%, 158.11% and 143.98%, respectively, NPQ of REX1, REX2 and REX3 lines increased by 25.88%, 61.41% and 65.81%, respectively. In general, the NPQ increase rate of OEX lines were significantly higher than that of NT and REX lines, i.e., REXs > NT > OEXs (*p* ≤ 0.05) (Figure 5E).

On the ninth day after salt stress, the photosynthetic light-response curves of Φ_PSII_ and ETR of NT, OEX and REX lines were drawn under 100 μmol·m^−2^·s^−1^, 200 μmol·m^−2^·s^−1^, 400 μmol·m^−2^·s^−1^, 600 μmol·m^−2^·s^−1^, 800 μmol·m^−2^·s^−1^ and 1200 μmol·m^−2^·s^−1^ light intensity, respectively. On the whole view, the Φ_PSII_ of NT and transgenic lines all decreased with the increase in light intensity. Compared with 100 μmol·m^−2^·s^−1^ light intensity treatment, the Φ_PSII_ of NT decreased by 70.34%, while the Φ_PSII_ of OEX1, OEX2 and OEX3 decreased by 87.30%, 72.22% and 68.71%, and the Φ_PSII_ of REX1, REX2 and REX3 decreased by 61.68%, 64.33% and 64.87%, under 1200 μmol·m^−2^·s^−1^ light intensity treatment (Figure 6A).

On the ninth day after salt stress, the ETR of NT increased with the increase in light intensity between 100 μmol·m^−2^·s^−1^ and 800 μmol·m^−2^·s^−1^, and finally slightly decreased when the light intensity increased to 1200 μmol·m^−2^·s^−1^. Meanwhile, the ETR of REX lines increased continually with the increase in light intensity. The ETR of OEX3 increased with the increase in light intensity from 100 μmol·m^−2^·s^−1^ to 800 μmol·m^−2^·s^−1^, and then reached a platform when the light intensity increased to 1200 μmol·m^−2^·s^−1^, the ETR of OEX1 and OEX2 increased first and then decreased under the light intensity range from 100 μmol·m^−2^·s^−1^ to 1200 μmol·m^−2^·s^−1^, and reached the maximum under 800 μmol·m^−2^·s^−1^ (Figure 6B).

### 2.6. General Information of RNA-seq on NT, OEX1 and REX1 Lines Under Salt Stress

After raw data screening, a total of 427,510,764 clean reads (min 25,018,818–max 31,138,582) were acquired from NT, OEX1 and REX1 apical bud and fifth functional leaf samples. The % ≥ Q30 value (the percentages of the bases whose quality score ≥ 30, namely, the base calling error rate ≤ 0.1%) of each sample was no less than 94.54%. According to alignment results, min 67.37%–max 73.45% clean reads can be mapped to *Populus trichocarpa* genome (Table 1).

### 2.7. The Differentially Expressed Genes (DEGs) Among NT, OEX1 and REX1 Lines Under Salt Stress

On the sixth day after salt stress treatment, 240 DEGs (167 up-regulated and 73 down-regulated DEGs) were found between OEX1 and NT apical buds, while 550 DEGs (363 up-regulated and 187 down-regulated DEGs) were found between REX1 and NT apical buds (Figure 7A, Appendix A). Meanwhile, there were 711 DEGs (366 up-regulated and 345 down-regulated DEGs) between OEX1 and NT fifth functional leaves, and 346 DEGs (126 up-regulated and 220 down-regulated DEGs) between REX1 and NT fifth functional leaves (Figure 7B, Appendix A). Additionally, among all the DEGs between OEX1 vs. NT and between REX1 vs. NT, in apical buds and fifth functional leaves, there were a lot of previously reported salt stress response TF genes such as *bHLH*, *WRKY*, *MYB*, *NAM* and *AP2/EREBP* genes (Appendix A).

### 2.8. GO Enrichment on the DEGs Among NT, OEX1 and REX1 Lines Under Salt Stress

GO (gene ontology) enrichment analysis indicate that the DEGs between OEX1 and NT apical buds were mainly related to cellular homeostasis process, metabolic process, oxidation reduction, iron ion and heme binding, and antioxidant activity, etc. (Appendix A). The DEGs between REX1 and NT apical buds were mainly related to microtubule-based process, chlorophyll biosynthetic and metabolic process, photosynthesis, polysaccharide and lipid metabolic process, etc. (Appendix A). The DEGs between OEX1 and NT fifth functional leaves were mainly enriched into aromatic amino acid family biosynthetic and metabolic process, chorismite and dicarboxylic acid metabolic process, and organic acid, small molecule, and fatty acid biosynthetic process, etc. (Appendix A). While the DEGs between OEX1 and NT fifth functional leaves were mainly enriched into lipid/cellular glucan metabolic process, photosynthesis, and organic acid, small molecule, and fatty acid biosynthetic process, etc. (Appendix A).

### 2.9. KEGG Enrichment on the DEGs Among NT, OEX1 and REX1 Lines Under Salt Stress

According to KEGG enrichment analysis, the DEGs between OEX1 and NT apical buds were mainly involved in plant hormone signal transduction, phenylpropanoid biosynthesis, phenylalanine metabolism, flavonoid biosynthesis, amino sugar and nucleotide sugar metabolism, etc. (Figure 8A). The DEGs between REX1 and NT apical buds were mainly related to DNA replication, flavonoid and phenylpropanoid biosynthesis, phenylalanine and purine metabolism, and photosynthesis, etc. (Figure 8B). The DEGs between OEX1 and NT fifth functional leaves were mainly involved in flavonoid and phenylpropanoid biosynthesis, phenylalanine, tyrosine and tryptophan biosynthesis, phenylalanine metabolism, glycolysis/gluconeogenesis, and plant hormone signal transduction, etc. (Figure 8C). The DEGs between REX1 and NT fifth functional leaves were mainly related to photosynthesis, flavonoid biosynthesis, nitrogen metabolism, and glycolysis/gluconeogenesis, etc. (Figure 8D).

### 2.10. Photosynthesis-Related DEGs Among NT, OEX1 and REX1 Under Salt Stress

According to the annotation information of all DEGs, the number of photosynthesis-related genes were differentially expressed among NT, OEX1 and REX1 apical bud and fifth functional leaf samples under salt stress. There were 13 DEGs (such as *Potri.013G137700*, ATP synthase subunit c, chloroplastic; *Potri.010G131800*, Phosphoenolpyruvate carboxylase 4; and *Potri.T063100*, Ribulose bisphosphate carboxylase large chain) were closely related to photosynthesis or chloroplast between OEX1 and NT apical buds. Meanwhile, there were 37 DEGs (such as *Potri.005G154500*, Photosystem II reaction center protein K; *Potri.019G067900*, Probable acyl-activating enzyme 6; and *Potri.019G093400*, Probable carotenoid cleavage dioxygenase 4, chloroplastic) closely related to photosynthesis or chloroplast between REX1 and NT apical buds. There were 53 DEGs (such as *Potri.015G043400*, Heterodimeric geranylgeranyl pyrophosphate synthase small subunit, chloroplastic; *Potri.019G075400*, Pentatricopeptide repeat-containing protein At1g71460, chloroplastic; and *Potri.018G138700*, Protein kinase APK1B, chloroplastic) closely related to photosynthesis or chloroplast between OEX1 and NT fifth functional leaves. There were 41 DEGs (such as *Potri.013G138300*, Photosystem II protein D1; *Potri.003G065200*, Photosystem I P700 chlorophyll a apoprotein A1; *Potri.009G044700*, Pentatricopeptide repeat-containing protein At2g29760, chloroplastic) closely related to photosynthesis or chloroplast between REX1 and NT fifth functional leaves (Appendix A).

### 2.11. QRT-PCR Verification of RNA-seq

According to the RNA-seq results, *PsnWRKY70* (Potri.016G137900) was significantly up-regulated in the apical bud (4.43 folds) and fifth functional leaf (6.79 folds) of OEX1 compared with NT under salt stress. Additionally, two photosynthesis-related genes, *AAE6* (33.86 folds, Probable acyl-activating enzyme 6, Potri.019G067900) and *CCD4* (26.04 folds, Probable carotenoid cleavage dioxygenase 4, Potri.019G093400) were also significantly up-regulated in REX1 apical bud compared with NT under salt stress. Meanwhile the photosynthesis-related genes, *PRPA* (0.05 folds, Pentatricopeptide repeat-containing protein At2g29760, Potri.009G044700) and *CPD11* (0.06 folds, Chaperone protein dnaJ 11, Potri.004G172300) in REX1, and *GPSSS* (0.06 folds, Heterodimeric geranylgeranyl pyrophosphate synthase small subunit, Potri.015G043400) in OEX1 fifth functional leaves were significantly down-regulated compared with NT under salt stress. And the present qRT-PCR verification demonstrated that the expression of *PsnWRKY70* (Potri.016G137900) were indeed higher in OEX1 apical bud than that in NT. And the expression of *AAE6* (Potri.019G067900) and *CCD4* (Potri.019G093400) were also significantly up-regulated in REX1 apical bud compared with NT (Figure 9A–C). Meanwhile, the expression of *PRPA* (Potri.009G044700) and *CPD11* (Potri.004G172300) in REX1, and *GPSSS* (Potri.015G043400) in OEX1 fifth functional leaves were also significantly lower than that in NT under salt stress (Figure 9D–F).

## 3. Discussion

Salt stress can damage plant cell membrane structure and function, intracellular ion balance and normal physiological metabolic activities [50]. Na^+^/K^+^ is an index that reflects salt damage degree and osmotic regulation capacity of plant [51]. In the present study, the Na^+^ and K^+^ content of NT, OEX1 and REX1 lines (both in leaf and root) were all maintained at a stable value under normal growth conditions. After 6 days of salt stress treatment, the Na^+^ content and Na^+^/K^+^ of three lines all significantly increased, while the K^+^ content of three lines all significantly decreased, both in leaf and root tissues. The Na^+^/K^+^ of OEX1 was much larger than that of NT and REX1 under salt stress. These results indicates that NT, OEX1 and REX1 all start resistance mechanisms by a higher Na^+^ accumulation in vacuoles and an increased Na^+^ extrusion in roots on the sixth day after salt stress treatment (120 mM NaCl). The much lower Na^+^/K^+^ of REX1 after 6 days of NaCl treatment indicates that it performed better than NT and OEX1 in resistance to salt stress.

SOD and POD are important antioxidases that reflect the oxidation resistance of plant. When plants experience salt stress, the activity of SOD and POD will increase to help adapt to environment change [52]. MDA content is an index to evaluate the membrane lipid peroxidation of plant under salt stress [53]. In this study, the SOD and POD activity increase rate of REXs was significantly larger than that of NT and OEXs, while the MDA content increase rate of OEXs was significantly larger than that of NT and REXs, under salt stress. These results indicate that REXs exhibited better osmotic regulation capacity than NT and OEXs, and OEXs suffered the most severe membrane lipid damage among all the lines.

As the fundamental way for plants to survive, photosynthesis not only produces carbohydrates for plants, but also provides essential oxygen and food source for the entire biosphere. When plants suffer salt stress, their chloroplast structure, chlorophyll content, chlorophyll fluorescence parameters and photosynthesis parameters will exhibit corresponding changes to help plants survive from stress damages [54,55]. Chlorophyll plays a key role in photosynthesis. It is a necessary element to absorb, transfer and convert light energy, especially in the light reaction stage. The content of chlorophyll directly affects the efficiency of photosynthesis, thus affecting the growth, development and yield of plants. In the present study, the chlorophyll content and CCI value of REX lines both exhibited a slightly greater decreasing trend than NT and OEX lines under salt stress. This indicates that the suppression of *PsnWRKY70* can help *P. simonii* × *P. nigra* to maintain chlorophyll (chloroplast) to survive from salt stress injury.

The net photosynthetic rate (P_n_) represents the actual amount of organic matter accumulated by plants in unit time. A higher P_n_ value means that plants use light energy more efficiently for the synthesis of organic matter, supporting faster growth and higher yields [44]. In our study, the P_n_ of NT and transgenic lines all significantly decreased 9 days after salt stress treatment. Compared with that of NT and OEX lines, the P_n_ of REX lines decreased most slightly. These results indicate that the suppression of *PsnWRKY70* can slow down the P_n_ decline of *P. simonii* × *P. nigra* under salt stress and maintain a relative stronger photosynthesis ability.

Stomatal conductance (G_s_) is an important index to measure the gas permeability of plant stomatal. A higher G_s_ means more CO_2_ will be absorbed by plant through their stomatal, thus promoting photosynthesis. Transpiration regulates water loss through the opening and closing of stomatal, thus affecting the water state of leaves. When plants experience salt stress, the water absorption capacity of the roots is insufficient, the leaves sense the lack of water supply and close the stomatal to prevent dehydration of the leaves, which leads to a weakened or stopped photosynthesis due to the lack of carbon dioxide [56]. In the present study, the G_s_ and transpiration rate (T_r_) of NT, OEX and REX lines both continuously decreased with the extension of salt stress duration. And among all the seven lines, REX lines exhibited the greatest but a slight G_s_ and T_r_ decline. These results indicate that suppression of *PsnWRKY70* can limit stomatal conductance, thereby reduce transpiration rate and maintain cell permeability under salt stress.

Intercellular CO_2_ concentration (C_i_) is also a vital parameter of plant photosynthesis. Under normal growth conditions, when the G_s_ increases, the amount of CO_2_ absorbed by leaves from outside increases, and the C_i_ increases, thus resulting in an increased photosynthetic rate. But when plants experience stress condition, the G_s_ decreases, the amount of CO_2_ absorbed by leaves from outside decreases, and the C_i_ always decreases first (caused by stomatal limitation) and then increases (caused by non-stomatal limitation), thus resulting in a decreased total photosynthetic rate [57,58]. In the present study, C_i_ of NT, OEX and REX lines first decreased and then increased with the extension of salt stress duration. C_i_ of NT and OEX lines reached the minimum on the third day after salt stress, while that of REX lines reached the minimum on the sixth day after salt stress. In other words, REX lines were under stomatal limitation during the first 6 days after salt stress, and then were under non- stomatal limitation. In contrast, NT and OEX lines were under stomatal limitation during just the first 3 days after salt stress, and then were under non-stomatal limitation. These results indicate that REX lines can maintain photosynthesis ability longer time than NT and OEX lines under salt stress.

Chlorophyll fluorescence parameters can reflect the photosynthetic physiological state of plants and can also be used to evaluate the degree of plant stress damage [59]. Photosynthetic electron transport rate (ETR) refers to the transfer speed of photosynthetic electrons in the photosynthetic electron transport chain in unit time, and its value is positively correlated with the strength of plant photosynthetic capacity. In the present study, the ETR of NT, OEXs and REXs all significantly decreased after 9 days of salt stress treatment, and the decent degree was ranked as OEXs > NT > REXs. This result indicates that REX lines maintain a better photosynthetic capacity than NT and OEX lines under salt stress.

The maximum photochemical efficiency (F_v_/F_m_), actual photochemical efficiency (Φ_PSII_) and photochemical quenching coefficient (q_P_) are important indicators that reflect the work capability of photosystem II. The F_v_/F_m_ of plants is usually relatively stable (0.80–0.84) in normal growth environment, and when plants are subjected to external environmental stress, F_v_/F_m_ will show a downward trend. In the present study, the decline trends of F_v_/F_m_, Φ_PSII_, and q_P_ of NT, OEX and REX lines on the ninth day after salt stress treatment were similar to that of ETR. These results indicate that salt stress inhibited the potential activity of photosystem II reaction center, reduced the efficiency of light energy conversion in photosystem II, and finally affected the function of photosynthetic organs. Among all the seven lines, the decent degree of F_v_/F_m_, Φ_PSII_, and q_P_ of REXs were the most slightly, and this indicates that the photosystem II of REX lines work better than NT and OEX lines under salt stress. The increase in non-photochemical quenching coefficient (NPQ) is usually related to plant’s photoprotection mechanism under stress conditions. The present study found that the NPQ increases the amplitude of seven lines, in order OEXs > NT > REXs, which indicates that the OEX lines suffered more severe salt stress damage than NT and REXs. Additionally, the subsequent Φ_PSII_ and ETR photosynthetic light-response curve analysis also indicates that REX lines exhibit better ability to activate the photosynthetic protection mechanism and adapt to a certain degree of strong light than NT and OEX lines under salt stress.

By analyzing the dynamic change trends of the phenotype, Na^+^ and K^+^ content, SOD and POD activity, MDA content, chlorophyll content, photosynthesis parameters and chlorophyll fluorescence parameters of NT, REX and OEX lines under salt stress with the extension of salt stress treatment duration, we found that REX lines exhibited a stronger osmotic adjustment ability, oxidation resistance capacity, and photosynthesis adaptability compared with NT and OEX lines under salt stress. In other words, the suppression of *PsnWRKY70* can help *P. simonii* × *P. nigra* to maintain a better photosynthesis and physiological state, and suffer less salt stress damage, while the overexpression of *PsnWRKY70* has the opposite effect, under salt stress. And this result is also in consistent with our previous study [49].

In order to further explore the molecular mechanism of the physiological and photosynthetic characteristic differences between transgenic and NT lines, RNA-seq analysis of the apical bud and fifth functional leaf of NT, REX1 and OEX1 after 6 days of salt stress was conducted. In general, the DEGs between REX1 vs. NT (apical bud and fifth functional leaf) and OEX1 vs. NT (apical bud and fifth functional leaf) were all different in amount and category. In the same tissue, the DEG difference was probably caused by the overexpression or suppression of *PsnWRKY70*. In the same line, the DEG difference was probably caused by tissue difference. Under salt stress, the expression of *PsnWRKY70* was significantly up-regulated in both the apical bud and fifth functional leaf of OEX1, and showed no significant change (namely maintained low expression level) in both the apical bud and fifth functional leaf of REX1. OEX1 plants were more salt sensitive, while REX1 plants were more salt tolerant, which indicates that *PsnWRKY70* plays a negative regulation role in *P. simonii* × *P. nigra* in response to salt stress. This is consistent with our previously reported study [49,60].

Among all the DEGs, photosynthesis-related DEGs in the same line were more enriched in the functional leaves than that in apical buds. This is probably due to the fact that the functional leaf suffered salt stress damage earlier than the apical bud, and the plant activates different photosynthetic protection mechanisms between apical bud and functional leaf under salt stress. In REX1, the expression of three *Photosystem I P700 chlorophyll a apoprotein A1* genes (Potri.003G065200, Potri.013G141800 and Potri.019G028100) and a *Photosystem II protein D1* gene (Potri.013G138300) were significantly up-regulated after 6 days of salt stress. This result indicates that the suppression of *PsnWRKY70* will probably enhance the photosynthesis adaptability of *P. simonii* × *P. nigra* by up-regulating these photosynthesis-related genes. In OEX1, the *Heterodimeric geranylgeranyl pyrophosphate synthase small subunit*, *chloroplastic* gene (Potri.015G043400) and *Phospho-2-dehydro-3-deoxyheptonate aldolase 1*, *chloroplastic* gene (Potri.007G095700) were significantly down-regulated after 6 days of salt stress. This indicates that the overexpression of *PsnWRKY70* probably weakens the photosynthesis adaptability of *P. simonii* × *P. nigra* by down-regulating these photosynthesis-related genes. However, these are just hypotheses, and we will further study the biological function and salt stress response signaling mechanism of these photosynthesis-related DEGs through a series of biological and biochemical experiments in the near future.

## 4. Materials and Methods

### 4.1. Plant Materials and Salt Stress Treatment

NT and transgenic (OEX1, OEX2, OEX3 and REX1, REX2, REX3 lines) *P. simonii* × *P. nigra* plants were grown in a phytotrone of State Key Laboratory of Tree Genetics and Breeding, Northeast Forestry University (Harbin, Heilongjiang, China) under the following growth conditions: day/night temperatures, 25 °C/20 °C; relative humidity, 50–60%; photoperiod, 16 h light/8 h dark; and photosynthetic photon flux, 400 μmol m^−2^ s^−1^. When the seedlings grew to approximately 30 cm, they were subjected to salt stress (120 mM NaCl). The apical buds and the fifth functional leaves of NT, REX1 and OEX1 (each sample contained tissues from at least three individual seedlings) were collected on the sixth day after salt stress treatment. After sample collection, the tissues were frozen immediately in liquid nitrogen and stored in −80 °C freezers for RNA extraction.

### 4.2. Na^+^ and K^+^ Ion Content Detection

We prepared 1–2 g fresh leaves or roots of NT, OEX1 and REX1, with a 105 °C kill out for 2 h, then dried (75 °C) the materials to a constant weight. We weighed the dry weight, added 25 mL aqua fortis and heated the sample to dissolve it. Cooling to room temperature, we filled the solution to 15 mL by deionized water. A full spectrum direct reading with an inductively coupled plasma emission spectrometer was performed for the detection (n = 5).

### 4.3. SOD, POD Activity and MDA Content Detection

SOD, POD activity and MDA content of NT, OEX1, OEX2, OEX3, REX1, REX2 and REX3 leaves under salt stress were detected by superoxide dismutase Assay kit-Visible Spectrophotometry (SOD-2-Y, Comin, Suzhou, China), peroxidase Assay kit-Visible Spectrophotometry (POD-2-Y, Comin, Suzhou, China) and Malonaldehyde Assay kit-Visible Spectrophotometry (MDA-2-Y, Comin, Suzhou, China), respectively (n = 5).

### 4.4. Chlorophyll Content and CCI Detection

After salt stress (0 d, 1 d, 3 d, 6 d, 9 d), the chlorophyll content index (CCI) value of NT, REX1, REX2, REX3, OEX1, OEX2 and OEX3 leaves was detected by a CCM-200 Plus chlorophyll content meter (OPTI-sciences, Tyngsboro, MA, USA). Then, we collected and cleaned the above-mentioned CCI-detected samples, and then added some quartz sand and 80% acetone to grind the samples and extract their chlorophyll. We measured the absorbance at 663 nm and 646 nm, and calculated the leaf chlorophyll concentration. We performed five biological replicates (n = 5) for each sample.

### 4.5. Photosynthesis Parameter Detection

After salt stress (0 d, 1 d, 3 d, 6 d, 9 d), the *P_n_*, *G_s_*, *T_r_*, and *C_i_* values of NT, REX1, REX2, REX3, OEX1, OEX2 and OEX3 seedlings were detected by a LI-6400XT portable photosynthesis measurer (Li-Cor Inc., Lincoln, NE, USA). Each sample contained five biological replicates (n = 5).

### 4.6. Chlorophyll Fluorescence Parameter Detection

At 0 d and 9 d after salt stress, a portable pulse modulated fluorometer FMS-2 (Hansatech, Kings Lynn, UK) was utilized to detect and calculate the chlorophyll fluorescence parameters of NT, REX1, REX2, REX3, OEX1, OEX2 and OEX3 seedlings. After 0.5 h dark adaption, minimal fluorescence (F_o_), maximal fluorescence (F_m_), potential photochemical activity of PSII (F_v_/F_o_), and F_v_/F_m_ of each sample were detected. Then, we opened the excitation light source (1800 μmol m^−2^ s^−1^), and detected the minimal fluorescence under light adaption (F’_o_), maximal fluorescence under light adaption (F’_m_), steady-state fluorescence under light adaption (F_s_) and maximal variable fluorescence under light adaption (F’_v_), and finally calculated Φ_PSII_, ETR, q_P_, and NPQ values of NT, OEX and REX lines. We performed five biological replicates (n = 5) for each sample. In addition, after 0.5 h dark adaption, the Φ_PSII_ and ETR photosynthetic light-response curve of each line was drawn by virtue of the built-in light source of FMS-2, with photon flux density (PFD) set at 100, 200, 400, 600, 800, and 1200 μmol m^−2^ s^−1^. The Φ_PSII_ and ETR values of each line under different PFDs were calculated according to the following formulas:Φ_PSII_ = (F_m_/F_s_)/F′_m_(1)
ETR = 0.5 × 0.85 × Φ_PSII_ × PFD(2)

### 4.7. RNA Extraction and Sequencing

Total RNA was isolated from the salt stress-treated apical bud and the fifth functional leaves of NT, REX1 and OEX1 using an EASYspin Plus Plant RNA kit (Aidlab, Beijing, China) according to the manufacturer’s instructions. Genomic DNA in the RNA extract was digested by DNase I (E1091, Omega Bio-Tek, Norcross, GA, USA). Then, the quality and quantity of total RNA were determined by Agilent 2100 bioanalyzer (Agilent Technologies, Palo Alto, CA, USA), Qubit 2.0 fluorometer (Invitrogen, Carlsbad, CA, USA), and NanoDrop 2000 spectrophotometer (Thermo Fisher Scientific, Waltham, MA, USA). Only the RNA samples with a 260/280 ratio between 1.8 and 2.0, a 260/230 ratio between 2.0 and 2.5, and a RIN (RNA integrity number) > 8.0 were used for sequencing. For each sample, an equal amount (approximately 20 μg) of total RNA was prepared for subsequent library construction and Illumina sequencing (Biomarker Technologies Co., Ltd., Beijing, China).

The sequencing libraries were constructed as described in the TruSeq RNA sample preparation guide (Illumina, San Diego, CA, USA). Then, the libraries were quantified using a Qubit 2.0 fluorometer and an Agilent 2100 bioanalyzer. Finally, the high-throughput Illumina HiSeq 4000 platform (Illumina, San Diego, CA, USA) was utilized for library sequencing (both ends of the inserts were sequenced).

### 4.8. RNA-seq Analysis and Verification

Artificial linkers, adaptors, and low-quality reads were removed from raw data to achieve high-quality clean reads. Then, the clean reads were mapped to the mRNA reference sequence in *Populus trichocarpa* (https://phytozome-next.jgi.doe.gov/info/Ptrichocarpa_v3_1, accessed on 1 October 2024) [61]. Transcript abundance was quantified using the Cufflink module in the Cufflinks program (ver. 2.1.1) [62]. Gene expression levels were calculated using fragments per kilobase of transcript per million mapped reads (FPKM) method in DEseq [63].

Differentially expressed genes (DEGs) were identified with a false discovery rate (FDR) < 0.05 (the *p* value was adjusted using the Benjamini–Hochberg method [64]) and fold change (FC) ≥ 2. DEGs were annotated in the GO and KEGG databases for sequence comparison and functional categorization. The gene ontology (GO) term analysis of DEGs was performed by the Blast2GO program. Enrichment analysis of GO terms and KEGG pathways was performed through the agriGO database (http://systemsbiology.cau.edu.cn/agriGOv2/FAQ.php?eqid=b9399b2a0003, accessed on 8 October 2024) and KOBAS (http://kobas.cbi.pku.edu.cn/program.run.do, accessed on 10 March 2024), respectively.

Finally, *PsnWRKY70* (Potri.016G137900) and several photosynthesis-related DEGs (Potri.019G067900, Potri.019G093400, Potri.009G044700, Potri.004G172300 and Potri.015G043400) were selected to verify the accuracy of RNA-seq by qRT-PCR. The specific primers were designed by Primer Premier 6.0 software and the Primer-BLAST tool in NCBI (http://www.ncbi.nlm.nih.gov/tools/primer-blast/index.cgi?LINK_LOC=BlastHome, accessed on 5 September 2024) (Appendix A), and the reference gene was *At4g33380-like* [32].

## 5. Conclusions

In order to compare the physiological and photosynthetic characteristics between *PsnWRKY70* transgenic and non-transgenic *P. simonii* × *P. nigra* lines, the Na^+^ and K^+^ content, superoxide dismutase (SOD) and peroxidase (POD) activity, malondialdehyde (MDA) content, chlorophyll content, photosynthesis parameters (P_n_, C_i_, G_s_ and T_r_) and chlorophyll fluorescence parameters (ETR, F_v_/F_m_, Φ_PSII_, q_P,_ NPQ and the photosynthetic light-response curves of Φ_PSII_ and ETR) of NT, OEXs and REXs under salt stress were detected. The results showed that REX plants exhibited better physiological photosynthetic characteristics under salt stress than NT and OEXs, indicating that the suppression of *PsnWRKY70* helps REXs to maintain a stronger osmotic adjustment ability, oxidation resistance capacity, and photosynthesis adaptability than NT and OEX lines under salt stress.

And then RNA-seq analysis of the apical bud and fifth functional leaf of NT, OEX1 and REX1 under salt stress was conducted. The results indicate that the DEGs between OEX1 vs. NT and REX1 vs. NT in different tissues (apical bud and fifth functional leaf) were all different in category and change trend. And there are a lot of salt stress response-related TF genes (such as *bHLH*, *WRKY*, *MYB*, *NAM* and *AP2/EREBP*) enriched in these DEGs. These indicate that the salt stress response mechanisms in the apical bud and fifth functional leaf of NT, OEX1 and REX1 plants were all different. On the sixth day after salt stress, the expression of *PsnWRKY70* was significantly up-regulated in both the apical bud and fifth functional leaf of OEX1, and showed no significant change (namely maintained low expression level) in both the apical bud and fifth functional leaf of REX1, thus indicating the negative regulation role of *PsnWRKY70* in *P. simonii* × *P. nigra* under salt stress.

Additionally, there were a lot of photosynthesis-related genes among all the DEGs in OEX1 vs. NT and REX1 vs. NT under salt stress. In REX1, the expression of three *Photosystem I P700 chlorophyll a apoprotein A1* genes (Potri.003G065200, Potri.013G141800 and Potri.019G028100) and a *Photosystem II protein D1* gene (Potri.013G138300) were significantly up-regulated after 6 days of salt stress, thus indicating that the suppression of *PsnWRKY70* probably enhances the photosynthesis adaptability of *P. simonii* × *P. nigra* under salt stress by up-regulating these photosynthesis-related genes. In OEX1, the *Heterodimeric geranylgeranyl pyrophosphate synthase small subunit* gene (Potri.015G043400) and *Phospho-2-dehydro-3-deoxyheptonate aldolase 1* gene (Potri.007G095700) were significantly down-regulated after 6 days of salt stress, thus indicating that the overexpression of *PsnWRKY70* probably weakens the photosynthesis adaptability of *P. simonii* × *P. nigra* under salt stress by down-regulating these photosynthesis-related genes. But this hypothesis should be verified by a series of biological and biochemical experiments through further study.

## Figures and Tables

**Figure 1 ijms-26-00081-f001:**
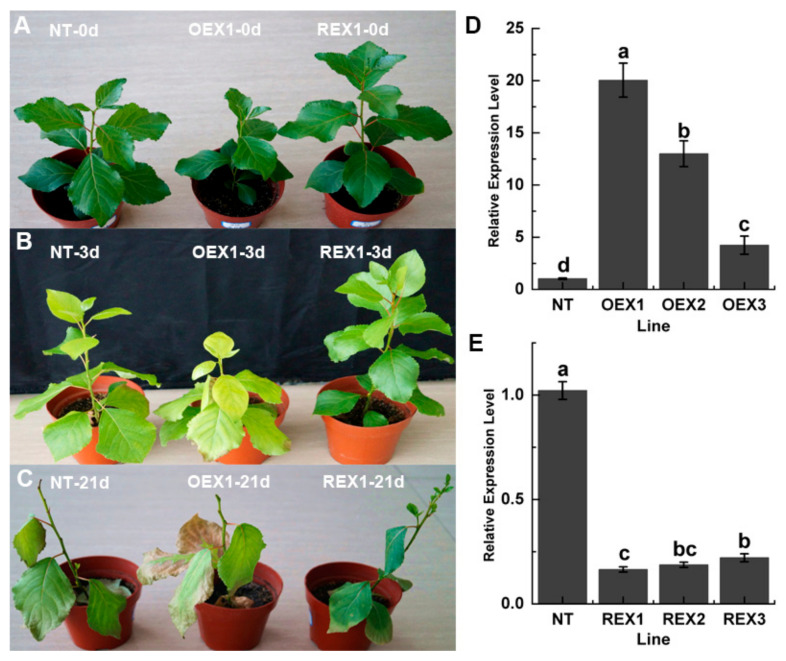
Phenotypic dynamic changes in NT, OEX1 and REX1 plants during salt stress treatment. (**A**) NT-0d, OEX1-0d and REX1-0d indicate NT, OEX1 and REX1 plants under normal growth conditions. (**B**) NT-3d, OEX1-3d and REX1-3d indicate NT, OEX1 and REX1 plants on the third day after salt stress treatment. (**C**) NT-21d, OEX1-21d and REX1-21d indicate NT, OEX1 and REX1 plants suffered 15 days of salt stress treatment and 6 days of rewatering. (**D**) The relative expression of *PsnWRKY70* in NT, OEX1, OEX2 and OEX3 lines. (**E**) The relative expression of *PsnWRKY70* in NT, REX1, REX2 and REX3 lines. Different letters on the columns indicate the significance of the difference (n = 3, *p* < 0.05, multiple-comparison method: Duncan).

**Figure 2 ijms-26-00081-f002:**
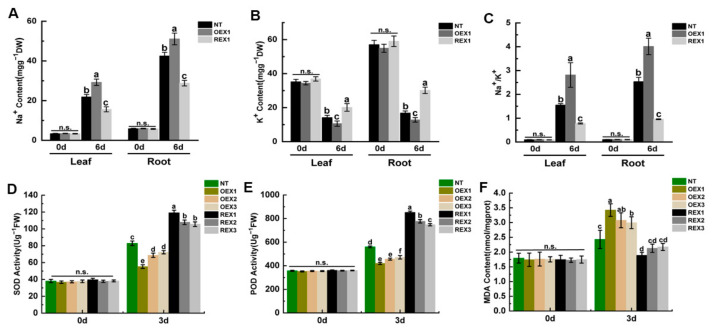
Na^+^, K^+^ content, SOD, POD activity and MDA content of NT, REX and OEX lines under salt stress. (**A**) Na^+^ content in the leaf and root tissues of NT, OEX1 and REX1 under salt stress. (**B**) K^+^ content in the leaf and root tissues of NT, OEX1 and REX1 under salt stress. (**C**) Na^+^/K^+^ in the leaf and root tissues of NT, OEX1 and REX1 under salt stress. (**D**) SOD activity of NT, OEX1, OEX2, OEX3, REX1, REX2 and REX3 under salt stress. (**E**) POD activity of NT, OEX1, OEX2, OEX3, REX1, REX2 and REX3 under salt stress. (**F**) MDA content of NT, OEX1, OEX2, OEX3, REX1, REX2 and REX3 under salt stress. Different letters on the columns indicate the significance of the difference (n = 5, *p* < 0.05, multiple-comparison method: Duncan), and “n.s.” means no significant difference.

**Figure 3 ijms-26-00081-f003:**
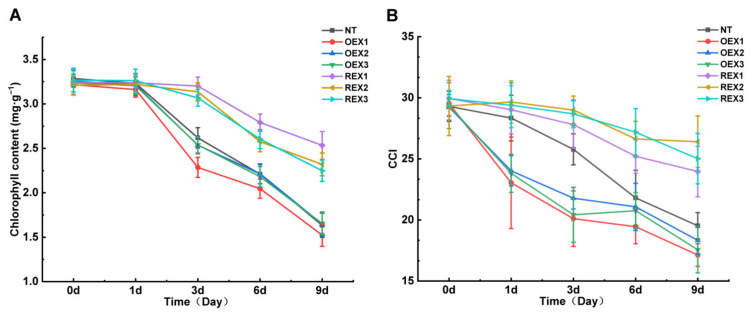
Chlorophyll content and chlorophyll content index (CCI) of NT, OEX and REX lines under salt stress. (**A**) The chlorophyll content dynamic change trends of NT, OEX1, OEX2, OEX3, REX1, REX2 and REX3 during salt stress treatment. (**B**) The CCI dynamic change trends of NT, OEX1, OEX2, OEX3, REX1, REX2 and REX3 during salt stress treatment.

**Figure 4 ijms-26-00081-f004:**
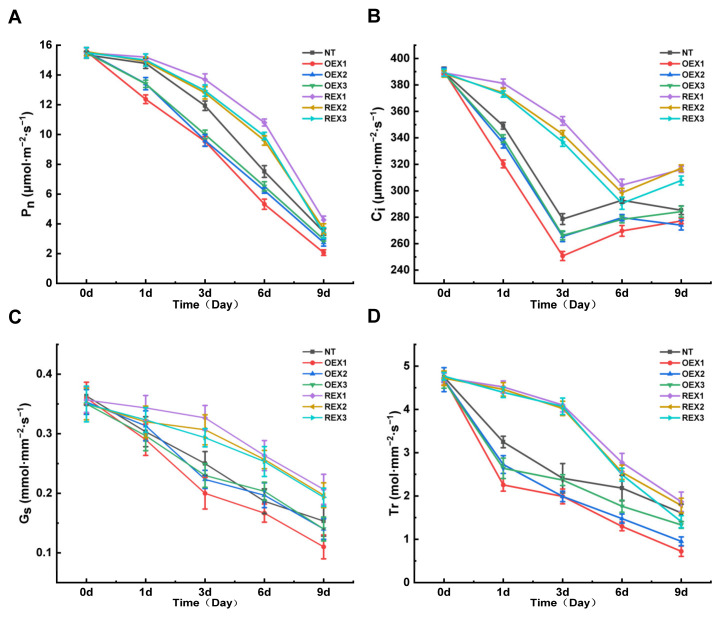
Photosynthesis parameters of NT, OEX and REX lines during salt stress. (**A**) The net photosynthetic rate (P_n_) dynamic change trends of NT, OEX1, OEX2, OEX3, REX1, REX2 and REX3 during salt stress treatment. (**B**) The intercellular CO_2_ concentration (C_i_) dynamic change trends of NT, OEX1, OEX2, OEX3, REX1, REX2 and REX3 during salt stress treatment. (**C**) The stomatal conductance (G_s_) dynamic change trends of NT, OEX1, OEX2, OEX3, REX1, REX2 and REX3 during salt stress treatment. (**D**) The transpiration rate (T_r_) dynamic change trends of NT, OEX1, OEX2, OEX3, REX1, REX2 and REX3 during salt stress treatment.

**Figure 5 ijms-26-00081-f005:**
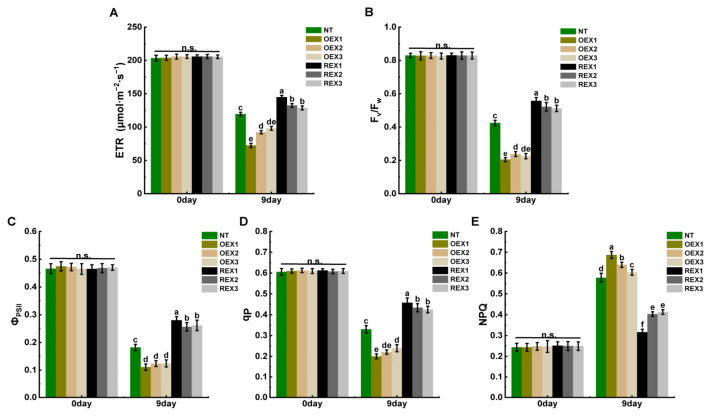
Chlorophyll fluorescence parameters of NT, REXs and NEXs lines before and after salt stress. (**A**) The photosynthetic electron transport rate (ETR) of NT, OEX1, OEX2, OEX3, REX1, REX2 and REX3 plants before and 9 days after salt stress treatment. (**B**) The maximum photochemical efficiency of photosystem II (PSII) (F_v_/F_m_) of NT, OEX1, OEX2, OEX3, REX1, REX2 and REX3 plants before and 9 days after salt stress treatment. (**C**) The actual efficiency of PSII (Φ_PSII_) of NT, OEX1, OEX2, OEX3, REX1, REX2 and REX3 plants before and 9 days after salt stress treatment. (**D**) The photochemical quenching coefficient (q_P_) of NT, OEX1, OEX2, OEX3, REX1, REX2 and REX3 plants before and 9 days after salt stress treatment. (**E**) The non-photochemical quenching (NPQ) of NT, OEX1, OEX2, OEX3, REX1, REX2 and REX3 plants before and 9 days after salt stress treatment. Different letters on the columns indicate the significance of the difference (n = 5, *p* < 0.05, multiple-comparison method: Duncan), and “n.s.” means no significant difference.

**Figure 6 ijms-26-00081-f006:**
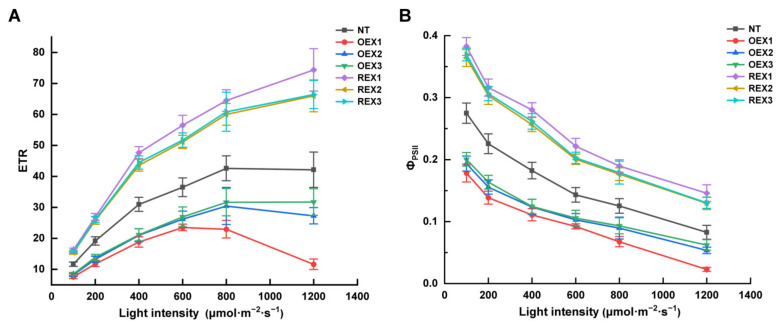
Photosynthetic light-response curves of Φ_PSII_ and ETR of NT, OEX and REX lines under salt stress. (**A**) Photosynthetic light-response curves of ETR of NT, OEX and REX lines under salt stress. (**B**) Photosynthetic light-response curves of Φ_PSII_ of NT, OEX and REX lines under salt stress.

**Figure 7 ijms-26-00081-f007:**
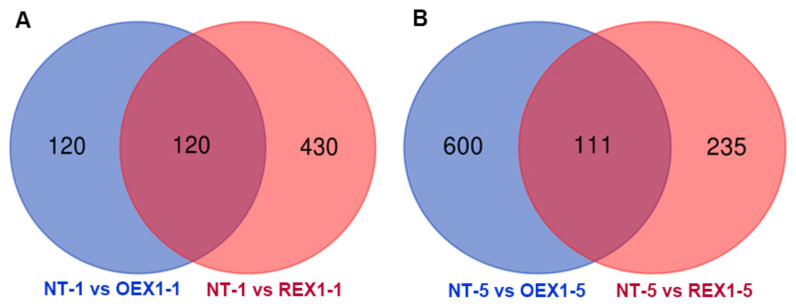
Venn diagram of the DEGs between OEX1 vs. NT and REX1 vs. NT 6 days after salt stress. (**A**) The DEGs between OEX1 vs. NT and REX1 vs. NT in their apical buds. NT-1, OEX1-1 and REX1-1 indicate the apical buds of NT, OEX1 and REX1 after 6 days of salt stress. (**B**) The DEGs between OEX1 vs. NT and REX1 vs. NT in their fifth functional leaves. NT-5, OEX1-5 and REX1-5 indicate the fifth functional leaves of NT, OEX1 and REX1 after 6 days of salt stress.

**Figure 8 ijms-26-00081-f008:**
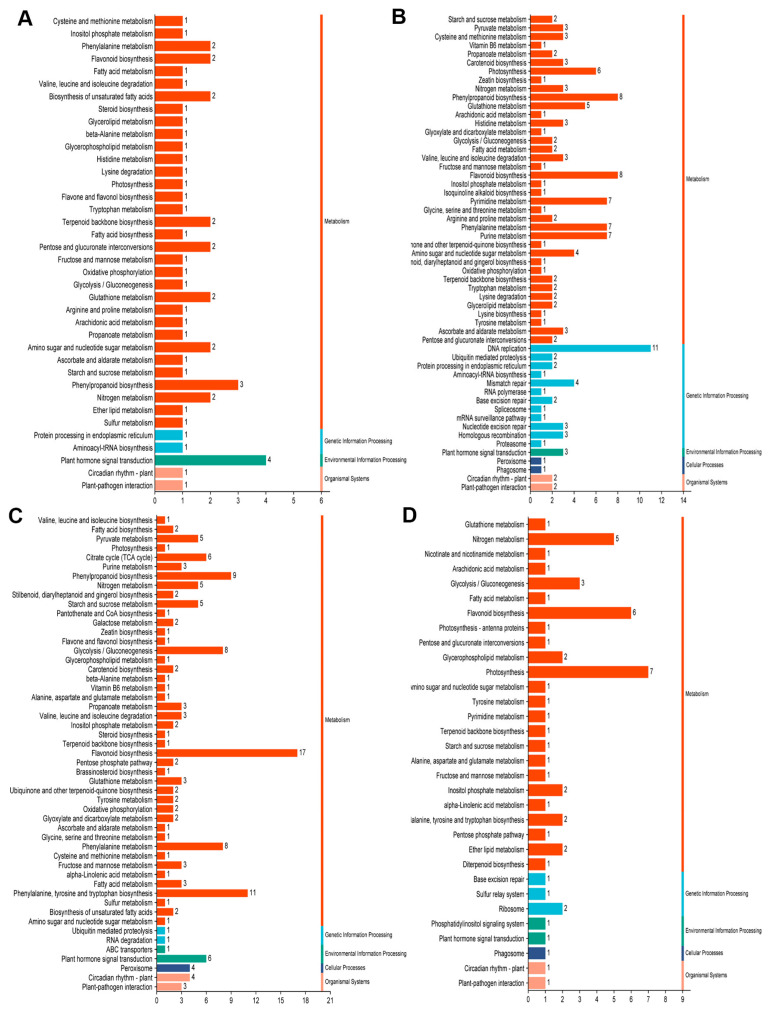
KEGG enrichment on the DEGs between OEX1 vs. NT and REX1 vs. NT 6 days after salt stress. (**A**) KEGG enrichment on the DEGs between OEX1 vs. NT apical buds. (**B**) KEGG enrichment on the DEGs between REX1 vs. NT apical buds. (**C**) KEGG enrichment on the DEGs between OEX1 vs. NT fifth functional leaves. (**D**) KEGG enrichment on the DEGs between REX1 vs. NT fifth functional leaves.

**Figure 9 ijms-26-00081-f009:**
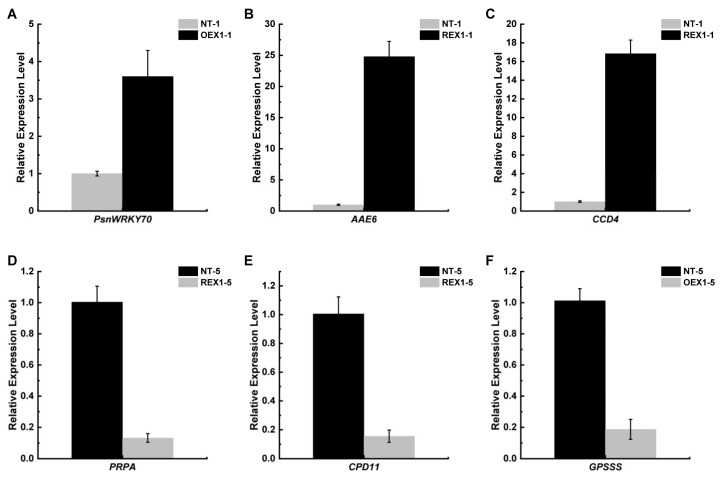
QRT-PCR verification on the expression change trends in several DEGs. (**A**) The expression level of *PsnWRKY70* (Potri.016G137900) in the apical buds of NT and OEX1 plants after 6 days of salt stress treatment. (**B**) The expression level of *AAE6* (Probable acyl-activating enzyme 6, Potri.019G067900) in the apical buds of NT and REX1 plants after 6 days of salt stress treatment. (**C**) The expression level of *CCD4* (Probable carotenoid cleavage dioxygenase 4, Potri.019G093400) in the apical buds of NT and REX1 plants after 6 days of salt stress treatment. (**D**) The expression level of *PRPA* (Pentatricopeptide repeat-containing protein At2g29760, Potri.009G044700) in the fifth functional leaves of NT and REX1 plants after 6 days of salt stress treatment. (**E**) The expression level of *CPD11* (Chaperone protein dnaJ 11, Potri.004G172300) in the fifth functional leaves of NT and REX1 plants after 6 days of salt stress treatment. (**F**) The expression level of *GPSSS* (Heterodimeric geranylgeranyl pyrophosphate synthase small subunit, Potri.015G043400) in the fifth functional leaves of NT and OEX1 plants after 6 days of salt stress treatment.

**Table 1 ijms-26-00081-t001:** General information of RNA-seq on NT, OEX1 and REX1 samples.

Sample Name	Clean Reads	Mapped Clean Reads	% ≥Q30
NT-1-01	25,483,638	17,248,446	95.26%
NT-1-02	26,888,658	18,763,764	95.48%
NT-1-03	25,328,278	17,320,518	94.92%
OEX1-1-01	25,018,818	17,138,316	95.13%
OEX1-1-02	25,208,248	17,596,887	94.54%
OEX1-1-03	25,066,578	16,888,285	95.16%
REX1-1-01	25,423,414	17,886,298	94.96%
REX1-1-02	26,285,570	18,344,311	95.82%
REX1-1-03	25,875,688	17,694,356	95.59%
NT-5-01	30,041,554	20,828,526	94.63%
NT-5-02	30,285,570	21,354,210	94.72%
NT-5-03	30,391,782	21,677,818	95.16%
OEX1-5-01	30,471,754	21,899,727	94.86%
OEX1-5-02	30,281,390	21,782,782	94.78%
OEX1-5-03	30,138,582	20,998,216	95.38%
REX1-5-01	30,972,826	22,310,078	95.46%
REX1-5-02	31,138,582	22,787,653	94.80%
REX1-5-03	30,910,408	22,702,314	95.27%

NT-1-01, NT-1-02, and NT-1-03 indicate the triple biological repetitions of NT apical bud samples on the sixth day after salt stress treatment; OEX1-1-01, OEX1-1-02, and OEX1-1-03 indicate the triple biological repetitions of OEX1 apical bud samples on the sixth day after salt stress treatment; REX1-1-01, REX1-1-02, and REX1-1-03 indicate the triple biological repetitions of REX1 apical bud samples on the sixth day after salt stress treatment; NT-5-01, NT-5-02, and NT-5-03 indicate the triple biological repetitions of NT fifth functional leaf samples on the sixth day after salt stress treatment; OEX1-5-01, OEX1-5-02 and OEX1-5-03 indicate the triple biological repetitions of OEX1 fifth functional leaf samples on the sixth day after salt stress treatment; REX1-5-01, REX1-5-02, and REX1-5-03 indicate the triple biological repetitions of REX1 fifth functional leaf samples on the sixth day after salt stress treatment. Mapped Clean Reads indicates the number of clean reads which can be mapped to *Populus trichocarpa* genome. % ≥ Q30, the percentages of the bases whose quality score ≥ 30 (i.e., the base calling error rate ≤ 0.1%).

## Data Availability

Data are contained within the article and Appendix A.

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
