# Peer review of "Physiological, Photosynthetic Characteristic and Transcriptome Analysis of PsnWRKY70 Transgenic Populus simonii × Populus nigra Under Salt Stress"

_ijms, 2024, doi:10.3390/ijms26010081_

Round 1
Reviewer 1 Report
Comments and Suggestions for Authors
The paper focuses on the behaviour of the photosynthetic parameters of several transgenic lines upon salt stress. The problem is that authors have made a great effort to construct those lines, but then most of the experiments do not seem appropriate to investigate salt stress tolerance. The major points:
a) Complete figure 1 with data on the expression of the transgene in each line, to check that the phenotypes correlate with the expression of the TF.
b) Authors have focused the research on the photosynthetic parameters, but this is only an indirect symptom of salt stress. With the experiments presented one cannot get information of how this TF factors is related to the salt stress response. For instance: Figure 7. The kegg categories cannot be read. But there are genes related to ABA signalling, osmotic response, biosythesis of osmolites or ion transport? This will give consistent information on the role of this gene upon salt stress.
c) Another missing experiments are: ion content (sodium and potassium) in roots and leaves, to determine whether the plants are tolerant by a higher accumulation of sodium in vacuoles or by an increased extrussion in roots.
Oxidation state: check, ROS, SOD, CAT and peroxidase activity and lipid damage to see if pants are withstaning better the effects (increased oxidativ response), or are suffering less stress (less lipid damage).
Author Response
Comment: Complete figure 1 with data on the expression of the transgene in each line, to check that the phenotypes correlate with the expression of the TF.
Response:First of all, we would like to express our appreciation for your valuable advices. According to your advice, the expression of the transgene in each line was added in Figure 1. The detailed correction was mainly presented in (Line 156-165) in the revised manuscript.
Reviewer 2 Report
Comments and Suggestions for Authors
Comments and Suggestions
I have an honor to review the manuscript entitled “Photosynthetic characteristic and transcriptome analysis on PsnWRKY70 transgenic Populus simonii × Populus nigra under salt stress” a research article submitted to the MDPI Journal, IJMS. Authors of this manuscript previously developed PsnWRKY70 overexpressed transgenic in populus. In this article they have analyzed differentially expressed genes among developed materials under salt stress and characterized genes using a series of bioinformatic and PRC analysis. Overall, the experiments they performed are well and the results are convincing. Thus, the presented results take up an important topic consistent with the profile of the Journal.
-However, even, manuscript is well organized and well described of the conception; I have some suggestions, which might improve the manuscript to make important to the wider audience.
-English should be improved throughout the text.
-Most suggestions I have mentioned in the main text pdf file. Please check
Title: -Good organization.
Abstract:
-Abstract is too long . Half of this is good enough. Need to write just core results. Many things you can remove from abstract like, 6 days after stress, 9 days after stress like redundant, less important subjects, methodology related, vague, redundant; like salt stress, P. simonii × P. nigra. This will reduce the volume of abstract.
1. Introduction
-Unnecessarily elongated, redundantly applied. Some description is less important, you may remove. Those distract the continuous flow of reading.
-L126-131; same information you wrote in the abstract. No need duplication.
-The aim of the study should be underlined precisely and simultaneously and highlight why this study is important. Rationale to be elucidated for the purpose of the study.
Results:
-So many time used “fifth functional leaf” “sixth day after salt stress treatment” “triple biological repetitions” apical bud- and some others. Those are unnecessarily elongate the length of article. You may reduce use of those.
Materials and Methods (M&M)
-Using “control” is better than non-transgenic or NT
-Even though it is from your previous experiment, you need to describe how did you produce transgenic, what was the gene sources, primers, how did you overexpressed, briefly describe with reference.
-In 4.1; you already mentioned sample for RNA. So further detailing not necessary. This kind of duplication you may remove from the whole text.
-Describe about qRT-PCR
Conclusion
-No need methodology here again.
-
Percent match: 31%
Should be below 20%

Author Response
Comment:Authors have focused the research on the photosynthetic parameters, but this is only an indirect symptom of salt stress. With the experiments presented one cannot get information of how this TF factors is related to the salt stress response. For instance: Figure 7. The kegg categories cannot be read. But there are genes related to ABA signalling, osmotic response, biosythesis of osmolites or ion transport? This will give consistent information on the role of this gene upon salt stress.
Response:Thank you very much for your valuable advice. According to your advice, we have added the SOD activity, POD activity, and MDA content detection data on NT, OEXs, and REXs under salt stress to provide more information of how this TF factors is related to the salt stress response, the detailed correction was mainly presented in (Line 14-52; Line 128-137; Line 166-196; Line 436-456; Line 532-554; Line 586-597; Line 665-681) in the revised manuscript.
As to the RNA-seq analysis data, we are sorry that provided poor pixel “Figure 7”, and we have redrawn it (renamed as “Figure 8”) in the revised manuscript and submit vectorgraph of KEGG enrichment result in .svg file to submission system. Among all the DEGs between NT vs. OEX1, and NT vs. REX1, most DEGs were related to plant hormone signal transduction, antioxidant activity, oxidation reduction, iron ion and heme binding, and photosynthesis etc. There were also some salt stress response TFs such as bHLH, WRKY, MYB and NAM. These information was presented and revised in “2.7” , “2.8”, and “2.9” in the revised manuscript (Line 334-384 ), Table S2 and Table S3.
Reviewer 3 Report
Comments and Suggestions for Authors
Dear authors, I am pleased to send you my review of your manuscript, with my suggestions and advice.

Author Response
Comment:Another missing experiments are: ion content (sodium and potassium) in roots and leaves, to determine whether the plants are tolerant by a higher accumulation of sodium in vacuoles or by an increased extrussion in roots.Oxidation state: check, ROS, SOD, CAT and peroxidase activity and lipid damage to see if pants are withstanding better the effects (increased oxidativ response), or are suffering less stress (less lipid damage).
Response:Thank you very much for your valuable advice. According to your advice, we have added the Na+ content, K+ content, Na+/K+, SOD activity, POD activity, and MDA content detection data on NT, OEXs, and REXs lines under salt stress, and related discussion was also added. The detailed correction was mainly presented in (Line 14-52; Line 128-137; Line 166-196; Line 436-456; Line 532-554; Line 586-597; Line 665-681) in the revised manuscript.
Round 2
Reviewer 1 Report
Comments and Suggestions for Authors
My major concern has been solved, so I can recommend publication.